# Anti-SARS-CoV-2 IgG Seroprevalence in Tyrol, Austria, among 28,768 Blood Donors between May 2022 and March 2023

**DOI:** 10.3390/vaccines12030284

**Published:** 2024-03-08

**Authors:** Anita Siller, Lisa Seekircher, Manfred Astl, Lena Tschiderer, Gregor A. Wachter, Julia Penz, Bernhard Pfeifer, Andreas Huber, Manfred Gaber, Harald Schennach, Peter Willeit

**Affiliations:** 1Central Institute for Blood Transfusion and Immunology, University Hospital Innsbruck, Tirol Kliniken GmbH, 6020 Innsbruck, Austria; anita.siller@tirol-kliniken.at (A.S.); manfred.astl@tirol-kliniken.at (M.A.); gregor.wachter@tirol-kliniken.at (G.A.W.); julia.penz@tirol-kliniken.at (J.P.); 2Institute of Health Economics, Medical University of Innsbruck, 6020 Innsbruck, Austria; lisa.seekircher@i-med.ac.at (L.S.); lena.tschiderer@i-med.ac.at (L.T.); 3Tyrolean Federal Institute for Integrated Care, Tirol Kliniken GmbH, 6020 Innsbruck, Austria; bernhard.pfeifer@tirol-kliniken.at (B.P.); andreas.huber@tirol-kliniken.at (A.H.); 4Division for Digital Health and Telemedicine, UMIT-Tirol, 6060 Hall in Tirol, Austria; 5Blood Donor Service Tyrol of the Austrian Red Cross, 6063 Rum, Austria; manfred.gaber@tirol-kliniken.at; 6Ignaz Semmelweis Institute, Interuniversity Institute for Infection Research, 1090 Vienna, Austria; 7Department of Public Health and Primary Care, University of Cambridge, Cambridge CB2 0BB, UK

**Keywords:** SARS-CoV-2 IgG antibodies, seroprevalence, Spike IgG antibodies, Nucleocapsid IgG antibodies, blood donors, COVID-19

## Abstract

Background: To provide updated estimates on SARS-CoV-2 antibody seroprevalence and average antibody titres for Central Europe. Methods: In repeat cross-sectional investigations (1 May 2022 to 9 March 2023) involving 28,768 blood donors in the Federal State of Tyrol, Austria (participation rate: 87.0%), we measured Spike receptor-binding domain (RBD) and Nucleocapsid IgG antibodies (37,065 and 12,645 samples), and estimated monthly seroprevalences and geometric mean titres. Results: Median age of participants was 45.4 years (range 18–70); 43.2% were female. Spike RBD IgG antibody seroprevalence was 96.3% (95% CI: 95.6–96.9%) in May 2022, 97.4% (96.7–98.0%) in December 2022, and 97.9% (96.4–98.8%) in March 2023. Among seropositive participants, geometric mean titres increased from 1400 BAU/mL (95% CI: 1333–1471) in May 2022 to 1821 BAU/mL (1717–1932) in December 2022, and dropped to 1559 BAU/mL (1405–1729) by March 2023. Furthermore, titres differed markedly by vaccination status and history of infection, with being the highest in participants with booster vaccination and prior infection. In autumn 2022, Nucleocapsid IgG antibody seroprevalence ranged from 36.5% (35.0–38.1) in September to 39.2% (37.2–41.2) in December 2022. Conclusion: Seroprevalence of SARS-CoV-2 antibodies in blood donors from Tyrol, Austria, was remarkably stable from May 2022 to March 2023. In contrast, average Spike RBD IgG antibody titres peaked in December 2022.

## 1. Introduction

Seroepidemiological studies are a key type of study in infectious disease epidemiology as they help quantify the proportion of a population with antibodies against a given pathogen (i.e., the seroprevalence), and may also determine the absolute level of these antibodies (i.e., the antibody titre), thereby providing useful information about disease immunity. In the case of severe acute respiratory syndrome coronavirus type 2 (SARS-CoV-2), seroepidemiological studies were particularly valuable to help explore differences by demographic groups, region, and across time in a specific population [1]. However, since the World Health Organisation (WHO) officially declared the end of the coronavirus disease 2019 (COVID-19) pandemic, up-to-date SARS-CoV-2 seroprevalence studies have become much sparser. As pointed out by the WHO-Unity Studies group, there should still be a continuous investment in serosurveillance studies to further monitor COVID-19 [1]. In addition to estimating seroprevalence, it is also crucial to quantify reliably the height of antibody titres in the population because the titres have been shown to correlate with the degree of protection from infection and, particularly, a severe course of disease [2].

Our study continuously assessed SARS-CoV-2 antibodies in more than 60,000 blood donors aged between 18 and 70 years in the Federal State of Tyrol in Austria and regularly reported results to local health authorities between June 2020 and March 2023. The present report extends our previously published data [3,4] by another 11 months from May 2022 to March 2023. Seropositivity was 95.8% (95% confidence interval [CI]: 94.9–96.4%) in April 2022 with a mean SARS-CoV-2 IgG antibody titre targeting the receptor-binding domain of the spike protein (Spike RBD IgG antibody) of 1437 Binding Antibody Units per millilitre (BAU/mL) (95% CI: 1360–1518) among seropositive blood donors in Tyrol, Austria [4]. Seropositivity in the Federal Region of Tyrol, at that time, was therefore very similar to the seroprevalence of 95.9% (95% CI: 92.3–97.8%) in March 2022 estimated across high-income European countries [1]. Since then, one smaller seroprevalence study was conducted in Salzburg, Austria, involving 2000 adults in the period of January 2022 to March 2023 [5]. Furthermore, in Austria, non-pharmaceutical interventions have been dropped, different Omicron subvariants have become dominant [6], and an increasing percentage of the population has acquired hybrid immunity, i.e., was exposed to SARS-CoV-2 infections and vaccinations. Although the WHO officially declared the end of the COVID-19 pandemic in May 2023, it is still important to know the latest populations’ immune status for public health policies, its differences in population groups, and the influence of prior infections and vaccinations.

The aim of this study was to give an update on SARS-CoV-2 IgG antibody seroprevalence and Spike RBD IgG antibody titres in Tyrol, Austria using the most recent data from our large-scale study on blood donors.

## 2. Materials and Methods

Details of the study design have already been described in previously published studies [3,4]. To be eligible for the study enrolment, participants needed to (i) be aged between 18 and 70 years, (ii) be permanent residents in Tyrol, and (iii) fulfil the general requirements for blood donors including being healthy. The present study reports on individuals who donated blood between 1 May 2022 and 9 March 2023. During the period, 294 different blood donation events took place in 178 different places covering all districts of Tyrol. In total, 28,768 participants out of 33,056 blood donors were enrolled in this study. The participation rate was 87.0%. For every blood donation, data on age, sex, and place of residence were collected routinely. Between September and December 2022, blood donors were further asked to complete a questionnaire concerning their history of SARS-CoV-2 infections and vaccinations. The questionnaire consisted of two different parts. In the first part, questions about infection history were addressed, including the total number of SARS-CoV-2 infections. If participants reported a prior infection, they were asked to report the date for each of the infections, if they had symptoms or were hospitalised. The second part of the questionnaire focused on the total number of COVID-19 vaccinations received. If participants reported to have received a COVID-19 vaccination in the past, they were asked to report month and year when it was administered.

Serum samples were drawn from blood donors, cooled at 4 °C, and processed within 30 h at the laboratory of the Central Institute for Blood Transfusion and Immunology of the University Hospital in Innsbruck, Austria. All samples were routinely tested for Spike RBD IgG antibodies using the quantitative Abbott SARS-CoV-2 IgG II chemiluminescent microparticle immunoassay (CMIA) (Abbott Ireland, Sligo, Ireland). Spike RBD IgG antibody values ≥7.1 BAU/mL were considered seropositive. A cut-off value for positivity of ≥7.1 BAU/mL has a sensitivity of 99.37% (95% CI: 96.50–99.97%) at ≥15 days after COVID-19 onset (post-symptom onset) and a specificity of 99.60% (99.22–99.80%) according to the manufacturer [7]. Between September and December 2022, samples were additionally assessed for SARS-CoV-2 IgG antibodies targeting the nucleocapsid protein (Nucleocapsid IgG antibody) using the qualitative Abbott SARS-CoV-2 IgG CMIA (Abbott Ireland, Sligo, Ireland). This assay has a sensitivity of 100% (95% CI: 95.89–100%) at ≥14 days after COVID-19 onset (post-symptom onset) and a specificity of 99.63% (99.05–99.90%), according to the manufacturer [7]. Both tests were performed on the Alinity i system (Abbott Ireland, Sligo, Ireland) according to the manufacturer’s instructions.

We quantified seroprevalence with Agresti–Coull 95% CIs [8] of Spike RBD IgG antibodies for each of the months between May 2022 and March 2023 and of Nucleocapsid IgG antibodies between September and December 2022. Among seropositive participants, we additionally summarised Spike RBD IgG antibody titres as geometric means and 95% CIs. Moreover, we compared monthly Spike RBD IgG antibody titres with a 7-day incidence of SARS-CoV-2 infections per 100,000 person-weeks and percentage vaccinated with one, two, three, and four doses in the population of the Federal State of Tyrol, Austria. Furthermore, we calculated the percentage of participants across different categories of Spike RBD IgG antibody titres (seronegative titres <7.1 BAU/mL and seropositive titres at <500, 500 to <1000, 1000 to <2000, 2000 to <3000, and ≥3000 BAU/mL). To allow cross-study comparisons, the categories were chosen in the same way as in the Shieldvacc-2 study [2], a study on Spike RBD IgG antibodies as a correlate of protection.

To investigate differences in Spike RBD IgG antibodies by population subgroups and thereby help identify potential gaps of immunity, we fitted multivariate regression models that included the variables age group (25 to <35, 35 to <45, 45 to <55, 55 to <65, 65 to 70 vs. <25), sex (male vs. female), and district (Innsbruck-Land, Schwaz, Innsbruck-Stadt, Kitzbühel, Imst, Landeck, Reutte, Lienz vs. Kufstein). Because we aimed to estimate the independent association of each of the three categorical variables, we entered them concomitantly in the same regression models and report associations of each variable adjusted for the other two. In particular, we used (i) a generalised estimating equation with a logit link function, a binomial distribution family, and an independent variance structure to test for differences in seroprevalence; and (ii) a linear mixed model with a random intercept to test for differences in antibody titres in seropositive participants. Heterogeneity was tested by the Wald test.

Among participants who completed the questionnaire between September and December 2022, we calculated seroprevalence and geometric means and 95% CIs of the most recent of Spike RBD IgG antibody titre measurements in population subgroups defined by different histories of SARS-CoV-2 infection (uninfected, latest infection before 2022, latest infection in 2022), vaccination status (unvaccinated, vaccinated without booster, vaccinated with booster [defined as having received three or four SARS-CoV-2 vaccinations], latest vaccination before 2022, latest vaccination in 2022), or different combinations of both. Furthermore, we used a linear regression model to test for differences in log-transformed Spike RBD IgG antibody titres across different combinations of history of SARS-CoV-2 infection (uninfected, infected) and vaccination status (unvaccinated, vaccinated without booster, vaccinated with booster) and additionally across combinations of vaccination status or history of SARS-CoV-2 infection with age and sex, with *p* values calculated from standard parametric Wald tests.

*p* values ≤ 0.05 were deemed as statistically significant and all statistical tests were two-sided. Analyses were carried out with Stata 15.1 and R 4.1.0.

## 3. Results

The results are reported in accordance with the Strengthening the Reporting of Observational Studies in Epidemiology (STROBE) guidelines (Appendix A).

Characteristics of the study participants are summarised in Table 1. In our study, 28,768 individuals were enrolled with a median age of 45.4 years (interquartile range [IQR] 31.1–55.4) and 12,440 being female (43.2%). Further details on the age- and sex-distribution of the study population are provided in Appendix A. We assessed Spike RBD IgG antibodies in all participants; of those, 21,716 (75.5%) had a measurement at one, 6281 (21.8%) at two, and 771 (2.7%) at three or more time points. Nucleocapsid IgG antibodies were assessed in 11,987 participants; of those, 11,587 (96.7%) had provided one, 266 (2.2%) two, and 134 (1.1%) three or more measurements.

Table 2 describes the evolution of SARS-CoV-2 IgG antibody seroprevalence during the course of our study. Seroprevalence of Spike RBD IgG antibodies was similarly high in each month with 96.3% (95% CI: 95.6–96.9%) in May 2022, 97.4% (96.7–98.0%) in December 2022, and 97.9% (96.4–98.8%) in March 2023. Seroprevalence of Nucleocapsid IgG antibodies was lower in September 2022 as compared to December 2022 with 36.5% (35.0–38.1%) and 39.2% (37.2–41.2%), respectively. Furthermore, we observed fluctuating Spike RBD IgG antibody titres during the course of our study.

Figure 1 depicts the geometric means of the Spike RBD IgG antibody titres per month with the SARS-CoV-2 incidence rate as well as the percentage of SARS-CoV-2 vaccinated in the Tyrolean population. Among seropositive participants, the geometric mean of Spike RBD IgG antibody titres decreased from 1400 BAU/mL (95% CI: 1333–1471) in May to 1093 BAU/mL (1047–1142) in July; afterwards, it increased to 1821 BAU/mL (1717–1932) in December 2022, and decreased again in the subsequent months to 1559 BAU/mL (1405–1729) in March 2023 (Table 2). This fluctuation is also noticeable in the percentages of participants across different categories of Spike RBD IgG antibody titres as shown in Appendix A. For example, 32%, 26%, 41%, and 32% of individuals belonged to the category with detectable titres at ≥3000 BAU/mL in May 2022, July 2022, December 2022, and March 2023, respectively.

Figure 2 summarises results investigating the cross-sectional correlates of Spike RBD IgG antibodies. Seropositivity was significantly higher in participants aged <25 years as compared to all age groups until <65 years (all *p* < 0.001) (Figure 2A). However, compared to this age group, the antibody titres were 8.6% (95% CI: 2.9–14.7%) and 19.5% (9.4–30.5%) higher in participants aged 55 to <65 years and ≥65 years, respectively, and 14.8% (9.8–19.5%) and 19.1% (14.3–23.6) lower in participants aged 25 to <35 years and 35 to <45 years, respectively (Figure 2B). Furthermore, Spike RBD IgG antibody titres and seroprevalence also differed across districts (*p* for heterogeneity <0.001). No differences in seroprevalence were detected by sex (*p* = 0.189) but antibody titres were 3.51% (95% CI: 0.21–6.92%) higher in males than in females. Table 3 summarises geometric means of Spike RBD IgG antibody titres of most recent measurements between September and December 2022 for different subgroups defined by prior SARS-CoV-2 infection and vaccination status. Spike RBD IgG antibody titres were the highest in participants with (i) the latest SARS-CoV-2 infection in 2022 (geometric mean 1946 BAU/mL [95% CI: 1875–2020]), (ii) the latest vaccination in 2022 (2507 BAU/mL [2398–2621]), and (iii) vaccination plus infection with booster vaccination (2758 BAU/mL [2683–2835]). Furthermore, we conducted an interaction analysis examining the differences in Spike RBD IgG antibody titre across groups defined by vaccination status and prior SARS-CoV-2 infection (Appendix A). On average, when comparing unvaccinated to booster-vaccinated participants, Spike RBD IgG antibody titres were 95% lower among those with a prior SARS-CoV-2 infection (95% CI: 94–95%) and 97% lower in those without a prior SARS-CoV-2 infection (94–99%) (*p* for interaction = 0.216). In addition, we did not observe a statistically significant interaction of vaccination status with age and sex (*p* = 0.283 and 0.593, Appendix A) nor of prior infection with sex (*p* = 0.947, Appendix A), but a significant interaction of prior infection with age driven by particularly high Spike RBD IgG antibody titres in the oldest participants (*p* < 0.001, Appendix A).

## 4. Discussion

Our study revealed that Spike RBD IgG antibody seroprevalence was 97.9% (95% CI: 96.4–98.8%) and the geometric mean antibody titre was 1559 BAU/mL (1405–1729), with 32% of the participants having an antibody titre of ≥3000 BAU/mL in March 2023. Seroprevalence was significantly higher in participants aged < 25 years compared to all other age groups, except the participants aged ≥ 65 years. A potential reason for this could be a higher number of social contacts in young age groups. An explanation for the high seroprevalence in participants aged ≥ 65 years would be a higher exposure to vaccinations, as vaccinations were highly recommended for older age groups. Furthermore, while there was no significant difference in seropositivity between sexes, among participants with detectable Spike RBD IgG antibodies, males had a slightly but significantly higher titre than females (+3.5%; *p* = 0.037), the reasons for which are unclear. As expected, participants that were vaccinated and had experienced a SARS-CoV-2 infection provided the highest Spike RBD IgG antibody titres if they had also received a booster vaccination.

The current analysis provides a much needed update on the populations’ immune status in Austria. Our last report on SARS-CoV-2 seroprevalence and antibody titres in Austria from April 2022, involved 22,607 blood donors and revealed a high seropositivity of 95.8% (95% CI: 94.9–96.4%) and a mean Spike RBD IgG antibody titre of 1437 BAU/mL (95% CI: 1360–1518) among seropositive blood donors from Tyrol [4]. Since then, another study, conducted in the Federal State of Salzburg, Austria, analysed antibody titres of 2000 participants between January 2022 and March 2023 [5]. Closely in line with our results, the study reported average Spike RBD IgG antibody titres of 3020 BAU/mL in participants who were vaccinated and convalescent, 771 BAU/mL in participants vaccinated only, and 66 BAU/mL in participants who were convalescent only [5]. Furthermore, as in our study, participants who had received three vaccine doses rather than two doses only had a significantly higher Spike RBD IgG antibody titre, with average levels of 835.5 vs. 361.5 BAU/mL, respectively [5]. Taken together, these data indicate clearly that antibody titres are highest in individuals that were infected and have received multiple doses of vaccines. All other prior population-based seroepidemiological studies in Austria, were conducted in 2021 before Omicron and its subvariants became predominant [3,9,10,11,12,13,14,15,16,17]. Since then, a large number of SARS-CoV-2 infections may have remained undetected due to (1) the decline of diagnostic testing, and (2) the growing number of asymptomatic infections [18,19,20]. Furthermore, a large proportion of individuals have developed hybrid immunity induced by a combination of SARS-CoV-2 infection and vaccination that has been presumed to afford a higher protection against infection and severe disease courses although sterile immunity cannot be achieved. In our present analysis, 53.4% of participants were vaccinated and have additionally gone through an infection.

Our data from Austria are in line with data from the Unity study showing a seroprevalence of 95.9% (95% CI: 92.3–97.8%) in March 2022 in high-income European countries [1]. Furthermore, another large-scale seroprevalence study that was conducted in Finland with 18 to 85 year olds from 2020 to 2022, reported Spike RBD IgG antibody titres of 97% (95% CI: 93–99%) from April to June 2022, 98% (76–100%) from July to September 2022, and 99% (97–99%) from October to December 2022, which are comparable to our results [21]. Our Spike RBD IgG antibody titres in May 2022 are further similar to a study from Switzerland including 2521 individuals, where seroprevalence was 93.8% (93.1–94.5%) in spring 2022 [22]. As in our study, no significant differences in seroprevalence were observed between men and women [22].

Dynamics of Spike RBD IgG antibody titres in our study population correlate with dynamics in incidence and vaccination rates of the Tyrolean population. When comparing the Spike RBD IgG antibody titres from our study period with the 7-day SARS-CoV-2 incidence per 100,000 person-weeks in the Tyrolean population, we observe, that a huge peak in 7-day incidence between September and November 2022 resulted in a peak of Spike RBD IgG antibody titres in December 2022. The slight delay of the Spike RBD IgG antibody peak in blood donors can be explained as blood donors are not allowed to donate blood directly after recovery when having gone through a SARS-CoV-2 infection knowingly. After recovery from a SARS-CoV-2 infection with or without fever >38 °C, blood donors were deferred from donation for 4 weeks or 1 week, respectively. Moreover, between September and December 2022, the percentage of the Tyrolean population vaccinated with four doses was rising constantly, which may also contribute to peaking Spike RBD IgG antibody titres in December 2022. In comparison, the percentage of seropositive Nucleocapsid IgG antibody blood donors only increased slightly from 36.5% (95% CI: 35.0–38.1%) in September 2022 to 41.0% (39.4–42.7) in November 2022. Potential explanations for this result could be that (i) a number of people do not have detectable Nucleocapsid IgG antibodies after being infected with COVID-19; (ii) waning of Nucleocapsid IgG titres within 2–8 months post-infection [23]; and (iii) people were deferred from blood donation for 4 weeks in case of a COVID-19 infection with fever >38 °C (potentially triggering a strong Nucleocapsid IgG antibody response) and for 1 week in case of a COVID-19 infection without fever. It also needs to be taken into account that individuals that have had a COVID-19 infection in October/November 2023, might not have returned to donate blood in December 2022 as they might not have felt well enough by then. Therefore, we assume that Nucleocapsid IgG antibodies were peaking in blood donors in January, where no data on Nucleocapsid IgG antibodies are available from our study. In comparison, there are other studies showing higher rates of Nucleocapsid IgG antibody seropositivity, potentially driven by cross-reactivity with other human corona viruses [24]. While the 7-day incidence was dropping between November and December 2022, the proportion of blood donors that were seropositive for Nucleocapsid IgG antibodies was stable and reached 39.2% (95% CI: 37.2–41.2) by December 2022. By comparing the percentage of participants in categories of Spike RBD IgG antibody titres from our previous study in April 2022 with the ones in March 2023, we can observe similar Spike RBD IgG antibody titres in spring 2022 and 2023. However, it needs to be considered when evaluating seroprevalences in a population, that Spike RBD and Nucleocapsid IgG antibody titres constantly decline after vaccination and/or infection, which was already shown by others [14,25,26] and in our previous study [3].

As expected from previous publications [27], when we compare unvaccinated and infected blood donors with vaccinated (with booster) and infected blood donors, we observe a higher seropositivity as well as higher antibody titres. This highlights that hybrid immunity leads to higher antibody titres but antibody titres also depend on whether individuals received a booster vaccination. For instance, the mean titres were similar in individuals that were vaccinated with a booster but uninfected and in individuals that were vaccinated without a booster but infected. Our data also show that in infected individuals, Spike RBD IgG antibody titres were higher in individuals with the latest infection in 2022 as compared to before 2022. Furthermore, antibody titres in individuals with the latest vaccination before 2022 were lower compared to titres in individuals with the latest vaccination in 2022. These results fit well with the results of other studies where protection against reinfection was shown to decrease when the time span to the last immunity-conferring event increases [28]. Concerning Spike RBD IgG antibody titres in different age groups, high seroprevalence in the youngest age group is comparable to the study from Finland, where seroprevalence was higher in younger people and especially high Nucleocapsid IgG antibody titres were detected [21].

The present study benefits from several strengths. First of all, involving 28,768 individuals, time-specific seropositivity as well as average Spike RBD IgG antibody titres were quantifiable in the whole study population as well as in several subgroups. Furthermore, our data are generalisable for the healthy Tyrolean population aged 18–70 years, as our blood donors represent a healthy subgroup of the Tyrolean population. Covering almost one year, our study is able to reflect antibody dynamics in Tyrol ranging from May 2022 to March 2023. Another strength is that we were able to discriminate between self-reported SARS-CoV-2 infection and vaccination or hybrid immunity in a subset of our study participants and estimated antibody titres as well as seropositivity in these relevant subgroups. However, this study has also some limitations. First, the main weakness of this study is that infection and vaccination status was self-reported as linking the data with the official data from the public health authority (infections and vaccinations) was not covered by the ethics committee approval. Second, the questionnaire enquiring infections/vaccinations of blood donors was only performed in a subgroup of 8457 blood donors. Third, we only measured Nucleocapsid IgG antibodies from September to December 2022 and not throughout the whole study period. Fourth, we did not include children/adolescents and people ≥70 years in our analysis, as blood donors only qualify for donation in Austria when being aged between 18 and 70 years. Therefore, we cannot provide an insight into seroprevalences and antibody titres in these population subgroups. Fifth, IgA or IgM antibody responses were not measured but would have been additionally valuable. Sixth, only Spike IgG antibodies directed against the RBD were measured.

## 5. Conclusions

In conclusion, seroprevalence of SARS-CoV-2 antibodies in blood donors from Tyrol, Austria, was remarkably stable from May 2022 to March 2023. In contrast, average Spike RBD IgG antibody titres peaked in December 2022. The updated results from this state-wide seroepidemiological study provide important insights into the SARS-CoV-2 antibody status of the general community and may thereby help inform public health policies and tailored interventions for the future.

## Figures and Tables

**Figure 1 vaccines-12-00284-f001:**
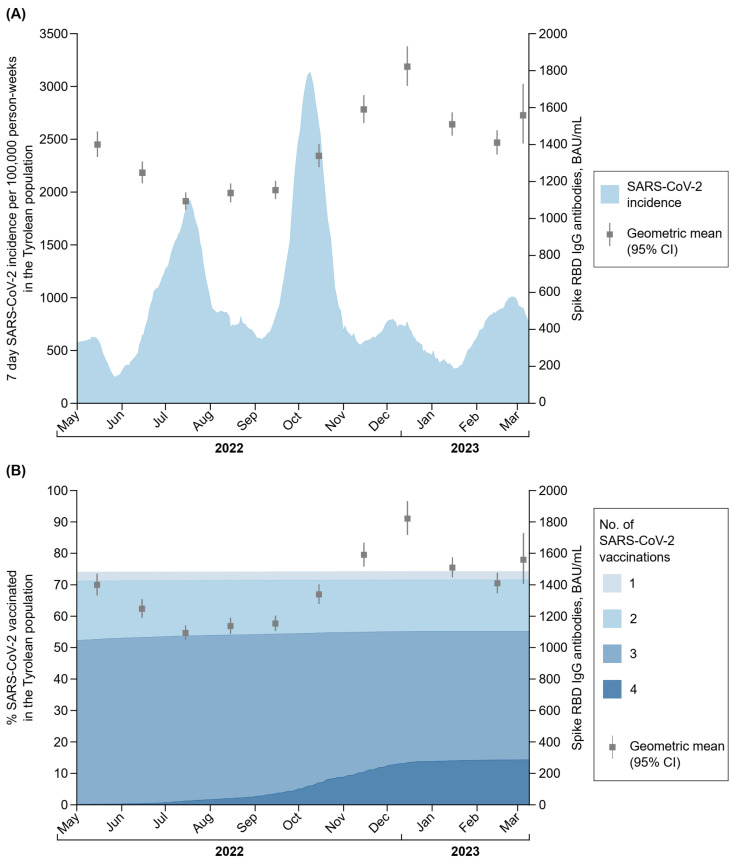
(**A**) Trajectory of Spike RBD IgG antibody titres and incidence rate in the Tyrolean population during the course of this study. (**B**) Trajectory of Spike RBD IgG antibody titres and % vaccinated in the Tyrolean population during the course of this study. The analysis involved data on 35,800 Spike RBD IgG antibody measurements taken from 27,845 seropositive participants. Data on SARS-CoV-2 infections and SARS-CoV-2 vaccinations in the Federal State of Tyrol, Austria, are publicly available at the AGES webpage. Abbreviations: BAU/mL, Binding Antibody Units per millilitre; CI, confidence interval; GM, geometric mean; RBD, receptor-binding domain.

**Figure 2 vaccines-12-00284-f002:**
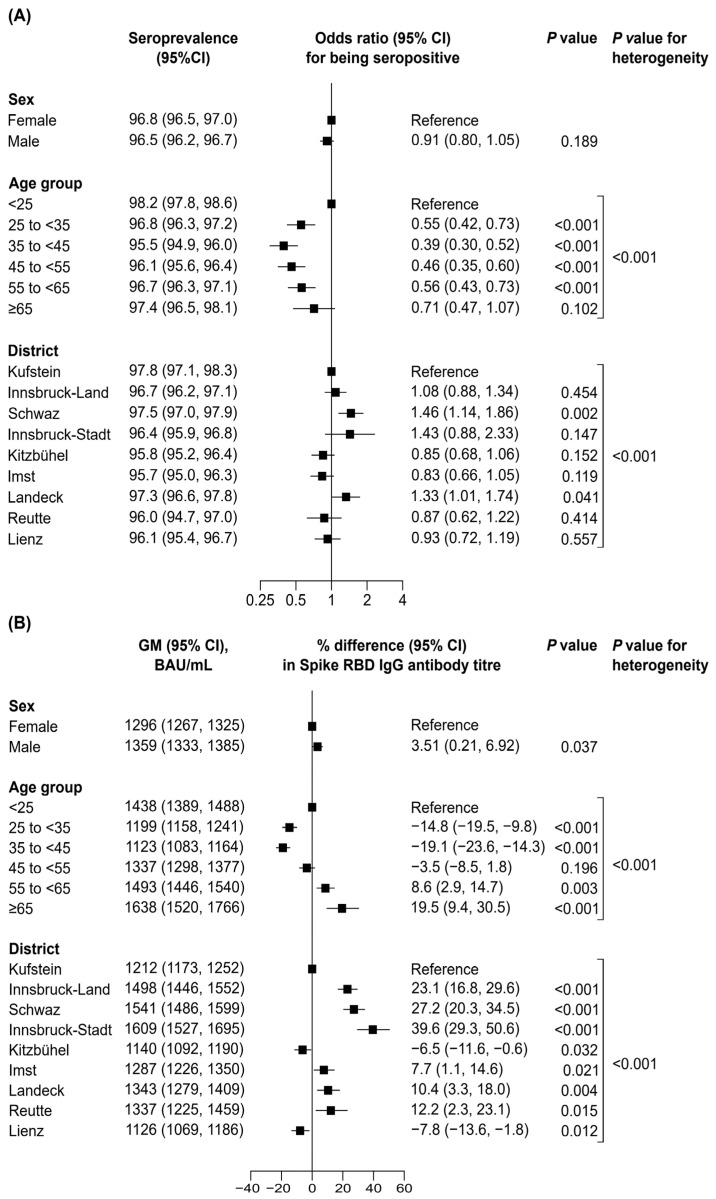
Cross-sectional correlation of sex, age, and district with seroprevalence (**A**) and antibody titre (**B**). Abbreviations: BAU/mL, Binding Antibody Units per millilitre; CI, confidence interval; GM, geometric mean. (**A**) The analysis involved data on 37,065 measurements taken from 28,768 individuals. To test for differences in seroprevalence, we used a generalised estimating equation with a logit link function, a binomial distribution family, an independent variance structure, and an adjustment for sex, age group, and district. (**B**) The analysis involved data on 35,800 seropositive measurements taken from 27,845 individuals. Seronegativity corresponds to Spike RBD IgG titres < 7.1 BAU/mL. To test for differences in antibody titres, we used a linear mixed model with a random intercept, and an adjustment for sex, age group, and district.

**Table 1 vaccines-12-00284-t001:** Characteristics of 28,768 participants enrolled in this study.

Characteristics	No. (%) or Median [IQR]
Age in years at baseline	45.4 [31.1–55.4]
Age groups at baseline	
<25	4125 (14.3%)
25 to <35	4951 (17.2%)
35 to <45	5063 (17.6%)
45 to <55	7128 (24.8%)
55 to <65	6401 (22.3%)
≥65	1100 (3.8%)
Sex	
Female	12,440 (43.2%)
Male	16,328 (56.8%)
Residence district at baseline	
Kufstein	5719 (19.9%)
Innsbruck-Land	5150 (17.9%)
Schwaz	4040 (14.0%)
Innsbruck-Stadt	1659 (5.8%)
Kitzbühel	3400 (11.8%)
Imst	2822 (9.8%)
Landeck	2296 (8.0%)
Reutte	1047 (3.6%)
Lienz	2635 (9.2%)
Availability of antibody measurements ^1^	
Spike RBD IgG antibodies	37,065 (100%)
Nucleocapsid IgG antibodies	12,645 (34.1%)

Abbreviations: IQR, interquartile range. ^1^ Percentage indicates proportion of total number of SARS-CoV-2 antibody measurements.

**Table 2 vaccines-12-00284-t002:** Seroprevalence and geometric mean of SARS-CoV-2 antibodies between May 2022 and March 2023.

		Spike RBD IgG Antibodies			Nucleocapsid IgG Antibodies	
Year	Month	N	% Seropositive (95% CI)	GM (95% CI) in BAU/mL ^1^	N	% Seropositive (95% CI)
2022	May	3314	96.3 (95.6–96.9)	1400 (1333–1471)	0	-
	June	3776	95.1 (94.4–95.7)	1248 (1190–1309)	0	-
	July	4615	95.7 (95.0–96.2)	1093 (1047–1142)	0	-
	August	3884	96.2 (95.5–96.7)	1137 (1087–1190)	0	-
	September	4142	96.6 (96.0–97.1)	1153 (1105–1204)	3641	36.5 (35.0–38.1)
	October	3421	97.3 (96.6–97.8)	1339 (1278–1403)	3394	39.6 (38.0–41.3)
	November	3399	97.4 (96.8–97.9)	1590 (1516–1668)	3382	41.0 (39.4–42.7)
	December	2247	97.4 (96.7–98.0)	1821 (1717–1932)	2228	39.2 (37.2–41.2)
2023	January	4432	96.8 (96.3–97.3)	1510 (1448–1574)	0	-
	February	3221	97.7 (97.1–98.2)	1410 (1346–1477)	0	-
	March	614	97.9 (96.4–98.8)	1559 (1405–1729)	0	-

The analysis involved data on 37,065 Spike RBD IgG antibody measurements taken from 28,768 individuals and 12,645 Nucleocapsid IgG antibody measurements from 11,987 individuals. Abbreviations: N, number of measurements; BAU/mL, Binding Antibody Units per millilitre; CI, confidence interval; GM, geometric mean; IQR, interquartile range. ^1^ Quantified among seropositive participants. Spike RBD IgG values are considered as positive if BAU/mL ≥ 7.1.

**Table 3 vaccines-12-00284-t003:** Spike RBD antibody titres of most recent measurements between September and December 2022 in individuals with information on SARS-CoV-2 vaccination and infection status.

	N	Seropositive, N (%) ^1^	GM (95% CI), BAU/mL ^2^
All	8457	8232 (97.3)	1537 (1491–1584)
Prior SARS-CoV-2 infection ^3^			
Uninfected	1764	1724 (97.73)	1067 (996–1143)
Latest infection before 2022	1425	1402 (98.39)	1019 (952–1091)
Latest infection in 2022	5268	5106 (96.92)	1946 (1875–2020)
SARS-CoV-2 vaccination status			
Unvaccinated	918	696 (75.82)	142 (127–159)
Vaccinated without booster ^4^	1433	1433 (100.00)	1063 (1007–1121)
Vaccinated with booster ^4^	6106	6104 (99.97)	2196 (2133–2261)
Latest vaccination before 2022	5129	5127 (99.96)	1687 (1634–1742)
Latest vaccination in 2022	2410	2409 (99.96)	2507 (2398–2621)
SARS-CoV-2 infection and vaccination status			
Unvaccinated + uninfected	45	7 (15.56)	31 (9–108)
Unvaccinated + infected	873	689 (78.92)	144 (129–161)
Vaccinated without booster ^4^ + uninfected	128	128 (100.00)	514 (400–662)
Vaccinated with booster ^4^ + uninfected	1591	1589 (99.87)	1150 (1072–1232)
Vaccinated without booster ^4^ + infected	1305	1304 (99.92)	1148 (1091–1208)
Vaccinated with booster ^4^ + infected	4515	4515 (100.00)	2758 (2683–2835)

Abbreviations: BAU/mL, Binding Antibody Units per millilitre; CI, confidence interval; GM, geometric mean; IQR, interquartile range. ^1^ Spike RBD IgG antibody values are considered as positive if BAU/mL ≥ 7.1. ^2^ Quantified among seropositive participants. ^3^ Refers to a SARS-CoV-2 infection prior to included Spike RBD IgG antibody measurement detected by self-report or by seropositivity of Nucleocapsid IgG antibodies. ^4^ Refers to three or four SARS-CoV-2 vaccinations.

## Data Availability

Data on SARS-CoV-2 infections and SARS-CoV-2 vaccinations in the Federal State of Tyrol, Austria are publicly available at the AGES webpage. Tabular data on the blood donor cohort can be requested from the corresponding authors by researchers who submit a methodologically sound proposal (including a statistical analysis plan); participant-level data on the blood donor cohort cannot be shared due to regulatory restrictions.

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
