# Peer review of "Anti-SARS-CoV-2 IgG Seroprevalence in Tyrol, Austria, among 28,768 Blood Donors between May 2022 and March 2023"

_vaccines, 2024, doi:10.3390/vaccines12030284_

Round 1

Reviewer 1 Report

Comments and Suggestions for Authors

This well-designed study describes the SARS-CoV-2 IgG Seroprevalence in Tyrol, Austria, among 28,768 Blood Donors between May 2022 and March 2023. Understanding the population seroprevalence to SARS-CoV2 is important to understand better COVID-19 and how best protect susceptible populations.

No issues were found with the methodology, results, or authors' conclusions.

Reviewer 2 Report

Comments and Suggestions for Authors

I have reviewed the provided file, which is a scientific article entitled " Anti-SARS-CoV-2 IgG Seroprevalence in Tyrol, Austria, among 28,768 Blood Donors between May 2022 and March 2023." The study offers important insights into the seroprevalence and antibody levels of anti-SARS-CoV-2 antibodies among blood donors in Tyrol, Austria. The study's large sample size and longitudinal design enhance the robustness of the results. The authors have adequately described the study design and methods, including participant eligibility criteria and data collection procedures.

Here are some comments:

Introduction:

- It would be helpful to provide a brief explanation of seroprevalence and antibody titers as some readers may not be familiar with these concepts.

- The study's findings in Tyrol, Austria, should be compared to those in other regions or countries with different vaccination and infection rates.

- It would be beneficial to include more details about the demographics of the study population, such as the distribution of age and sex among the participants. This would provide a better understanding of Tyrol's blood donor population.

Materials and Methods:

- The section provides a concise description of the study design and participant eligibility criteria. It mentions the enrolment period, the number of blood donation events, and the total number of participants. However, providing more specific information on the assays used to measure anti-SARS-CoV-2 antibodies would be helpful.

- It would be helpful to provide more context on the multivariable regression models used to investigate differences in anti-S IgG antibodies by population subgroups, including how these models account for potential confounding factors and the interpretation of the results.

- The questionnaire administered between September and December 2022 should be described in more detail, including the specific questions asked about SARS-CoV-2 infections and vaccinations and how the responses were used in the analysis, if possible.

Results:

- The results section presents key findings related to anti-Spike IgG seroprevalence and antibody titers among blood donors in Tyrol, Austria, from May 2022 to March 2023. The data are presented clearly, including the sample sizes, confidence intervals, and temporal trends. The differences in antibody titers based on vaccination status and history of infection are also highlighted. Overall, the results are well-organized and supported by the data presented.

- It would be helpful to include a legend for Table 1 that shows the total numbers of anti-S IgG and anti-N IgG. Also, including table and figure numbers in the document for easier reference would be beneficial. Additionally, data and numbers for males should be added to Table 1.

- The authors mention that the antibody titers differed significantly based on vaccination status and history of infection. It would be valuable to provide a more detailed analysis of these differences, including subgroup analyses and statistical comparisons, to elucidate the impact of vaccination and prior infection on antibody titers.

- Although the study primarily focuses on anti-Spike IgG antibodies, it briefly mentions the seroprevalence of anti-nucleocapsid IgG antibodies. It would be beneficial to provide more information and analysis on the trends and dynamics of anti-nucleocapsid IgG seroprevalence over time, if possible.

Conclusion:

- The conclusion provides a summary of the main findings regarding anti-SARS-CoV-2 antibody seroprevalence and titers among blood donors in Tyrol, Austria. However, it would be beneficial to include a final statement that emphasizes the significance of the findings and their implications for public health policies or interventions.

Reviewer 3 Report

Comments and Suggestions for Authors

1. Very little data was provided with the serological test for anti-nucleocapsid antibodies, which was tested only for a short period of a four months and did not show any significant changes at a time when the incidence of COVID-19 cases increased by 6-fold as shown in Figure 1.   

2. The anti-nucleocapsid test, is problematic, because more than half of people who get COVID-19 don’t have detectable nucleocapsid levels with standard commercial serological tests. They will get higher titre levels after multiple exposures to the SARS-CoV-2 virus. It would have been better if antibodies against other SARS-CoV-2 proteins besides the nucleocapsid and spike protein were tested for.  

3. The anti-spike antibody test used was focused on just the receptor binding domain region of the Spike protein, which is actually relatively poorly immunogenic and not a reliable marker of the degree of immune protection conferred by COVID-19 vaccines or prior infection. I know that the anti-SARS-CoV-2 antibody responses are extremely variable from person to person, although they are remarkably consistent over time for the same person.  

4. The presentation of the data in the tables is a bit confusing.  

5. The results are not unexpected, i.e., that antbody levels decline after infection, but can be restored after exposure to a vaccine that permits production of the spike protein. It is also not surprising that most people have SARS-CoV-2 spike antibodies, either from previous infection and/or vaccination. Two years into the COVID-19 pandemic, most people would have already been infected.  

6. Since a large portion of adults and especially children are asymptomatic for SARS-CoV-2 infections, even during their first infection, it is difficult to tell how many were already infected prior to and after the availability of COVID-19 vaccines. Thus I am not convinced that the conclusions are meaningful with respect to the benefits of vaccination after someone has had COVID-19.  

7. It is unfortunate that measurement of IgG antibodies in serum were the focus of the testing. This might be useful for monitoring antibody responses to a vaccine injected into the muscles. However, viral infection would be producing a much more robust IgA and IgM response that would be more meaningful from a protection from respiratory infection standpoint.

Reviewer 4 Report

Comments and Suggestions for Authors

The authors present the results of their seroprevalence study conducted using serum and medical history from blood donors, who represent a healthy subgroup of the Tyrolean population, and extend previously published data by 11 months, from May 2022 to March 2023. The aim of the study was to update knowledge on anti-SARS-CoV-2 IgG responses in the population. The authors follow the STROBE guidelines.

Major remarks

In the reviewer’s opinion Figures 1 and 2 convey the same message about the dynamics of the IgG titers, while supplementary Figure S2 contains important data that shows significant effects and is discussed in the last section of the article. Perhaps Figure S2 better deserves to be included in the main article than Figure 2.

Figure S2 (as discussed in lines 204-205) highlights that the middle-aged participants had lower seroprevalence and lower titers than the reference <25 years-old group. Did the authors analyze whether this was associated with a different vaccination history in these groups? Was it only age related?

Minor remarks

Lines 117-118 are somewhat misleading: “vaccinated without booster [defined as having received three or four SARS-CoV-2 vaccinations], vaccinated with booster” ; presumably it is the booster vaccination itself that the authors define as third and fourth vaccinations and not the without booster group.

What does the superscript d stand for in Table 3?

Round 2

Reviewer 3 Report

Comments and Suggestions for Authors

1.     Lines 40-45 – Change to “Seroepidemiological studies are a key type of study in infectious disease epidemiology as they quantify the proportion of a population with antibodies against a given pathogen (i.e., the seroprevalence), and may also determine the absolute level of these antibodies (i.e., the antibody titre), thereby providing useful information about disease immunity.” Antibodies are not developed against a disease, but the pathogen that causes the disease. Also “i.e.” is usually followed by a comma.

2.     Lines 99-and 100. It should be clarified that this test was developed to detect antibodies only against the receptor-binding domain region of the spike protein and does not access the vast majority of antibodies made against the spike protein. Consequently, while it might give a sense of “neutralizing” antibodies, it is not ideal for a serological test to assess overall spike protein-based immunity. It would be more appropriate to refer to “spike RBD antibody” rather than “spike antibody” throughout the manuscript. Referring to “anti-S IgG” antibody is actually misleading, because readers may think that this is the antibody level against the whole spike protein. It is also unfortunate that IgG antibody levels were specifically examined (which would be okay for an intra-muscular vaccination response) but not from a respiratory infection which would produce more IgA and IgM antibodies. I note that the authors commented on this in Line 346 and 347.

3.     Lines 106-108 – The authors have now added the manufacture’s claim of 100% sensitivity of detection of the nucleocapsid antibody. The question is whether everyone who has had COVID-19 has produced anti-nucleocapsid antibodies for determination of the degree of infectivity of the population. On the one hand, about half of the people that recover from an initial infection with SARS-CoV-2 and get COVID-19 do not have detectable nucleocapsid antibodies (now acknowledged in Lines 302 and 303).  On the other hand, about 90% of people in British Columbia had detectable antibodies against the spike or nucleocapsid protein using two different assays methods between May 15 and June 15 in 2020 (Majdoubi et al. (2021) A majority of uninfected adults show preexisting antibody reactivity against SARS-CoV-2. JCI Insight. 2021 Apr 22;6(8):e146316. doi: 10.1172/jci.insight.146316. PMID: 33720905).

4.     Lines 147 and 148 – The actual statistical test used for the p value determinations should be described. Was this, for example, a Student t-test or A-Nova?

5.     Lines 155 and 156 – Since a significant portion of the participants had been vaccinated more than once, it would have been interesting and important to know how each group (differentiated by number of vaccine doses) fared in terms of whether they were vaccinated after they had COVID-19 and whether they got COVID-19 subsequently, and what kind of Spike antibody levels they had. Unfortunately, it would appear that very few participants  in this study were not vaccinated (so there are few proper controls to evaluate the effectiveness of the vaccines) and very few (~3.3%) were monitored for anti-nucleocapsid antibodies if they had two or more doses (so there is no data on the effectiveness of multiple vaccinations). Studies such as those conducted by the Cleveland Clinic showed reduced efficacy of COVID-19 vaccines for protection from COVID-19 with increasing vaccination (Shrestha et al. (2023) Effectiveness of the Coronavirus Disease 2019 Bivalent Vaccine. Open Forum Infectious Diseases. 10(6): ofad209. Open Forum Infectious Diseases, Volume 10, Issue 6, June 2023, ofad209, https://doi.org/10.1093/ofid/ofad209). It seems to me that the authors should be able to determine this with the kind of data that they have collected.

6.     Lines 163-169 – The authors still have not really explained why the anti-nucleocapsid antibody level increased from September to December by at best 12% if at all, and yet the number of COVID-19 cases had increased according to Figure 1A by 6-fold. This is a big red flag that the anti-nucleocapsid antibody detection is not reflecting COVID-19 cases.

7.     Table 2. I find it fascinating that the anti-spike RBD antibody levels were all high at around 96 to 98% positive detection in the study participants from May 2022 to March 2023, yet there was a large increase in the number of detectable COVID-19 cases in the Tyrolean general population. Obviously, the vaccines were not effect in preventing further COVID-19 cases. The data that should really have been provided was the incidence of COVID-19 amongst the participants that were actually in the study.

8.     Lines 213-218 – This sentence is confusing, especially since it is unclear at the end where it states “and 97% lower in those without…” Without what? I presume without a previous infection, but this should be explicitly stated. The implication from this table is that infection and booster vaccination together gives the highest antibody titers. Yet, it is unclear that this reduced the incidence of COVID-19 in the participants. The vast majority of participants in the study had an infection in 2022 (62.2%).

9.     Table 3 - It seems that the anti-spike RBD antibody levels in those that were double vaccinated and supposedly uninfected was half of what was measured for those that were either boosted (but not infected or were double vaccinated and infected. A striking almost 2.4-fold increase in anti-spike RBD antibody titre was evident with infected people that also got one or more booster shots. This is remarkable, because the test for the measurement for anti-spike RBD antibody titre is based on the original Wuhan strain. The booster with the Omicron B1/B5 mutations was not available during the study.  Infection with Delta and Omicron variants clearly increased anti-spike RBD domain antibodies, with which the vast majority of people fully recovered from their infection. So clearly their immune responses were sufficient to neutralize the virus.  From this study, the Wuhan-based vaccines also further increased anti-spike RBD antibodies that targeted specifically the earlier SARS-CoV-2 variants that were extinct during the study period. One would have to concluded that despite the high levels of antibodies against the SARS-CoV-2 spike protein in highly vaccinated individuals this was poorly effective at preventing someone from getting COVID-19. The data is not presented in a manner that refutes this.

10.  Lines 253-257 – The point of this study was to monitor immune status against SARS-CoV-2 by either previous natural infection and/or COVID-19 vaccination. Much of the study was conducted during the periods with the Delta, Omicron B1 and Omicron B4/B5 waves. Antibody levels would be expected to be elevated by both the high prevalence of COVID-19 during this period and the fact that a population was highly vaccinated was studied. Even those that were vaccinated probably had sustained SARS-CoV-2 antibody levels due to exposure of the virus in the environment, even though they may not have developed COVID-19 symptoms. Consequently, it is hard to evaluate whether the so called “uninfected” individuals that were boosted were in fact infected but asymptomatic. The higher levels of anti-spike RBD antibodies in the boosted and infected probably is a reflection of the higher need for more antibodies to control infections were advance enough to cause disease and overcome it. The failure of this study is to not more carefully correlate the antibody levels with the propensity to develop COVID-19. With respect the antibody levels in general against SARS-CoV-2 following infection, this will be expected to decline to much lower levels in the absence of on going exposures to the virus as the population develops increasing natural immunity. This natural immunity will be a reflection of the levels of memory B and T cells that recognize the virus.

11.  Over all, I am supportive of publishing this major study. However, I would encourage the authors to present better data that correlates immune protection as monitored with serological antibody levels to the risk of getting COVID-19 disease.  Obviously, comparisons cannot be sufficiently made with unvaccinated individuals, but some differentiation could be made relative to vaccination status by dose.
